# Differential Expression and Diagnostic Value of MUC5AC Glycoforms in Pancreatic Ductal Adenocarcinoma

**DOI:** 10.3390/cancers15194832

**Published:** 2023-10-02

**Authors:** Ashish Manne, Lianbo Yu, Phil A Hart, Allan Tsung, Ashwini Esnakula

**Affiliations:** 1Department of Internal Medicine, Division of Medical Oncology at the Arthur G. James Cancer Hospital and Richard J. Solove Research Institute, The Ohio State University Comprehensive Cancer Center, Columbus, OH 43210, USA; 2Department of Biomedical Informatics, The Ohio State University, Columbus, OH 43210, USA; 3Division of Gastroenterology, Hepatology, and Nutrition, The Ohio State University Wexner Medical Center, Columbus, OH 43210, USA; 4Department of Surgery, University of Virginia Health, Charlottesville, VA 22908, USA; 5Department of Pathology, The Ohio State University Wexner Medical Center, Columbus, OH 43210, USA

**Keywords:** pancreatic cancer, mucin, MUC5AC, 45M1, CLH2, pancreatic ductal adenocarcinoma, diagnostic marker, prognostic marker

## Abstract

**Simple Summary:**

MUC5AC is a mucin with unclear importance in pancreatic cancers. Recent studies suggest the existence of two major classes of this mucin, which differ in location and function. Prior studies did not clearly establish the role of both kinds of MUC5AC in pancreatic cancer. We attempted to study their differential expression in non-cancer and cancer tissues and also tried to establish their impact in cancer tissues.

**Abstract:**

We explored the differential expression and diagnostic value of two significant Mucin 5AC (MUC5AC) glycoforms, less-glycosylated immature (IM) and heavily-glycosylated mature (MM), in neoplastic diseases (NpD), including pancreatic ductal adenocarcinoma (PDA) and neuroendocrine tumors (NET), and non-neoplastic (non-NpD) diseases. Commercially available tissue microarray (TMA) was constructed from 96 patients, including 38 primary PDA (PT), 5 metastatic lesions (ML), 11 NET, and the rest being non-NpD tissues. Immunohistochemistry for MUC5AC was performed using CHL2 and 45M1 clones for IM and MM isoforms, respectively. MUC5AC (both glycoforms) are not detected in non-NpD. In MUC5AC-positive neoplastic tissues, IM was localized to the cytoplasm (Cy) while MM was identified in apical (Ap) and extracellular (Ec) regions too. One ML positive (omentum) in the TMA expressed both. For PDA vs. non-PDA, the sensitivity (SN) was higher with MM ± IM (71%) than MM (47%) or IM (65%)-alone. The specificity (SP) was 100% with MM-alone, which dropped with the addition of IM (96%) or IM-alone (93%). For NpD vs. non-NpD, the SN (MM + IM-59%, IM-55%, MM-37%) was inferior, and SP was 100% for both glycoforms (MM ± IM). The combination of MUC5AC glycoforms has high SP and reasonable SN to diagnose PDA. They have the potential to be a reliable diagnostic marker and should be investigated further in more extensive studies.

## 1. Introduction

Pancreatic ductal adenocarcinoma (PDA) is a lethal disease with dismal outcomes [1]. The 5-year survival rate is just over 12% and accounted for approximately fifty thousand cancer-related deaths in the United States in 2022. Diagnosis in advanced stages, limited treatment options, and aggressive biology are key factors for poor survival [2]. A typical PDA patient presents with abdominal pain or jaundice (depending on the location of the primary, head vs. body and tail) [3]. Currently, imaging is the only modality for PDA screening, and the screening is limited to high-risk populations with known specific inherited genetic syndromes or a history of familial pancreatic cancer and is not recommended for the general population [4]. We extensively discussed the need for reliable and accessible biomarkers for PDA in our previous publications [5,6,7,8]. We focused on mucin 5AC (MUC5AC) as a biomarker in our publications.

MUC5AC can be divided into two major glycoforms, mature MUC5AC (MM) and immature MUC5AC (IM), based on the extent of glycosylation [6,7,8,9]. The MM is heavily glycosylated and is primarily identified at apical (Ap) or extracellular (Ec) regions of tumor cells and can be detected using specific monoclonal antibodies (mab) such as 45M1, Nd2, 1-13M1, and 21M1 using immunohistochemistry (IHC). The IM is present in the cytoplasm (Cy) or perinuclear and can be detected using CLH2 mab via IHC. The IM expression-based studies could distinguish abnormal pancreatic tissues, including PDA, mucinous cystic neoplasms, intraductal papillary mucinous neoplasm (IPMN), and metaplastic pancreatic duct epithelium from normal pancreatic tissues but did not give conclusive results on its predictive and prognostic value [6,10]. As we argued in our recent reviews, the impact of MUC5AC expression on PDA outcomes is unclear as the published studies did not explore the clinical significance of MM as a potential diagnostic, prognostic (survival or aggressive clinicopathological characteristics), or predictive biomarker (treatment response) [6,11]. There is strong preclinical evidence suggesting MM offers gemcitabine resistance to PDA; it was first reported by Krishn et al. but escaped attention as none of the prior studies were designed to examine it [12]. The authors showed that gemcitabine sensitivity improved with the inhibition of MUC5AC in the pancreatic cell lines. We elaborated on it in a paper published earlier this year [7]. Similarly, the relationship between IM and MM, and its effect on carcinogenesis, local invasion, metastasis, and treatment response was not investigated.

We aimed to examine the differential expression of MM and IM in neoplastic diseases (NpD) such as PDA and neuroendocrine tumor (NET) patients and non-neoplastic diseases (non-NpD) using tissue microarrays (TMA). We assessed the diagnostic value of MM and IM, individually and in combination, to distinguish NpD from non-NpD and PDA from non-PDA. This study is in line with prior TMA based studies that studied IM expression in various pancreatic disease groups, but for the first time the focus was on the differential expression (MM vs. IM) of the glycoforms for diagnosis and prognosis in PDA [11,13].

## 2. Materials and Methods

### 2.1. Specimen

A commercially available pancreatic cancer progression TMA (PA2081c; tissuearray.com, MD, USA) was used in the study. The TMA consisted of tissue from 96 patients with two 1 mm cores from each patient’s tissue. The TMA had 54 cases of neoplastic tissue with 43 PDA (38 primary (PT), 5 metastatic lesions (ML)) and 11 NET (including 10 islet cell tumors). Forty-two cases of non-NpD, including 10 normal pancreatic parenchyma, 24 normal pancreatic tissue adjacent to neoplasm, and 8 cases of pancreatitis (1 acute and 7 chronic), were included in the TMA (Table 1 below).

### 2.2. Staining

IHC was performed on TMA sections using a CHL2 clone (ab77576, Abcam, Boston, MA) for the IM isoform and a 45M1 clone (ab369, Abcam, Boston, MA, USA) for the MM isoform. Prior preclinical studies that showed MUC5AC’s role in cell proliferation, invasion, metastasis, and gemcitabine resistance used 45M1 clone, and, hence, it was used to examine MM expression [12]. The immunostains were optimized per the manufacturer’s recommendations using appropriate controls of normal gastric tissue. Briefly, following deparaffinization and rehydration of the TMA sections, heat-induced epitope retrieval was performed using a citrate-based antigen unmasking solution at pH 6.0 (H-3300-250, Vector Laboratories, Newark, CA, USA). A primary antibody for 45M1 and CHL2 clones and a secondary antibody were used at a dilution of 1:100. Primary antibody detection was carried out using a Horseradish Peroxidase-based detection system with conjugated rabbit anti-mouse-secondary-antibody at 1:100 dilution. The staining development was achieved via incubation with 3,30-diaminobenzadine (DAB) and DAB Enhancer (SK-4100, Vector Laboratories, Newark, CA, USA).

Immunohistochemically stained sections were scored by a pathologist (AE). Individual tissue cores were scored for the intensity of reactivity (0, no staining; 1+, weak staining; 2+, moderately intense staining; and 3+, strong staining) and the percentage of reactive tumor cells. H-score was derived from the product of these measurements resulting in a value between 0 and 300 [14]. Cases were categorized as having negative/weak (score < 10) or positive (score > 10) expression. An individual case was considered positive if at least one of the two cores had ≥10% of cells with any intensity. For every positive case, the pattern of expression in terms of apical (Ap), cytoplasmic (Cy), and extracellular (EC) expression was recorded.

### 2.3. Statistics

Descriptive statistics was provided for data summarization. Sensitivity (SN), specificity (SP), positive predictive value (PPV), negative predictive value (NPV), and accuracy were calculated with their respective formulas. Fisher’s exact test was used to test differences between groups. T-test was used to compare the means of CLH2-Hscore in MM positive and negative tumors. Software JMP 20 (SAS, Carry, NC, USA) was used for data analysis.

## 3. Results

### 3.1. Baseline Characteristics

The breakdown of the patients in the TMA is described in Table 1. The population was divided in two major ways to mimic the typical approach in the clinic. The first way was NpD vs. non-NpD. The NpD includes PDA (PT and ML) and NET. The non-NpD group includes normal parenchyma, normal tissue adjacent to tumor, and pancreatitis. The second way was dividing the population into PDA and non-PDA (NET and non-NpD). The PDA group had tissues from 38 PTs of which 14, 22, and 2 cases were from patients with clinical stage I, II, and III, respectively. Most PTs were conventional adenocarcinomas (n = 33) with variant histology in five patients (mucinous carcinoma-1, undifferentiated carcinoma-1, and adenosquamous carcinoma-3). Two PDAs were reported as invading the small intestine. One of the five MLs was from omentum, while the rest were from distant lesions (three liver and one unspecified soft tissue).

### 3.2. Distribution and Differential Expression of MUC5AC Glycoforms in Pancreatic Tissues

Non-NpD tissues did not express MUC5AC glycoforms (Figure 1). Two NETs (islet cell tumors) were positive only for IM, and 65% (28/43) of PDAs (27 PT and 1ML) were positive for MM or IM. The breakdown of the staining is detailed in Table 1.

The staining pattern was very distinct between two glycoforms when positive (Figure 2); MM showed Cy, Ap, and Ec expression, while IM was noted only in the cytoplasm (Cy). Membranous staining was negative for both mabs. Mean H-score for MM and IM was 24 and 60, respectively, in PDA (PT + ML). Mean H-score for CLH2 was significantly higher in MM-positive than MM-negative PDA (120 vs. 24, *p* = 0.0003).

Unlike CLH2 staining (always Cy when positive), the staining pattern of 45M1 clone is variable among PDAs (Figure 3 and Figure 4). Various combinations of Ap, Cy, and Ec were noted (described in detail below, Table 1, and Appendix A).

### 3.3. Distribution among PDAs

In PTs, the detection rates of MM ± IM, MM-alone, or IM-alone in PTs are 71% (27/38), 24% (9/38), and 5% (2/38), respectively. Both glycoforms (MM + IM) were expressed in 42% (16/38), while 29% (11/38) did not express both (Table 1). MM here refers to MM positivity irrespective of the location (Ap vs. Cy vs. Ec). None of the stage III patients on the TMA expressed MUC5AC (MM or IM), while 57% (8/14) of stage I and 86% (19/22) of stage II patients expressed one of them (MM ± IM). The detailed breakdown of expression in PTs is discussed in Appendix A. In omental metastatic lesions of the TMA, both MM (Ap and Cy, no Ec) and IM were detected. In liver and soft tissue lesions, MUC5AC was not detected.

### 3.4. Differential Expression of MUC5AC Glycoforms in PDA

The relation between MM detected at different places within a tumor cell (Ap vs. Cy vs. Ec) and between IM and MM in PT was studied (Appendix A). Two PDA lesions expressed MM without IM. Alternatively, 9 tumors expressed IM without MM. Ec MM detection was always associated with MM at other locations (Cy or Ap MM) or IM. About half (6/11) with Ec and 8/11 with Cy MM had a simultaneous expression of Ap MM. Three patients with Ap MM (1 had Ec, too) did not have detectable Cy MM. Various combinations of Ap, Cy, and Ec MM detection with respect to IM expression were noted (Appendix A). All PDAs with Ap + Ec or Cy + Ec combinations also had IM expression, while 1/8 of patients with the Ap + Cy MM combination did not express IM. Limited positivity among ML (1/5) in the TMA did not allow us to establish any specific patterns among ML. As reported above, omental lesions expressed AP + Cy MM and IM.

### 3.5. MUC5AC Expression and Pathological Characteristics in PDA

The pathological features available to us (stage, histology, grade of differentiation, node positivity, and size of the lesions) in PT with respect to MUC5AC expression was examined (Appendix A). The MUC5AC (MM ± IM) and IM expressions were significantly different (*p* < 0.05) and higher in early-stage (I–II) than late stage (III–IV) tumors. Stage II tumors had a higher fraction of positive cases. Other expression patters such as MM or IM-alone, Cy, Ap, and Ec were not significantly different among the different stages of PDA. One mucinous PDA patient expressed MM at all three locations (Ec, Ap, and Cy) and IM. MM was not expressed in undifferentiated and adenosquamous variants. One patient with adenosquamous variant expressed IM. One of the two patients with small intestine invasion expressed MM (EC, Ap, Cy), and both expressed IM.

Ap MM expression was significantly (*p* = 0.02) higher in well-to-moderately (G1–G2) differentiated tumors (10/22, 45%) than poorly (G3) differentiated (1/12, 8%) and undifferentiated and adenosquamous variants (0/4, 0%). Cy or Ec MM and IM were not significantly different among the groups, but the positive fractions were numerically more in G1–G2 than the other two groups. If we compare the G1–G2 (n = 22) group with the rest of the tumors (G3 + UC + AC, n = 16), any MUC5AC (MM ± IM) expression was significantly (*p* = 0.03) more in the former group. Even though MUC5AC (MM ± IM) was expressed more in node-positive than node-negative samples (88% vs. 67%) and T1–T2 than in T3–T4 (70% vs. 65%), the difference is not statistically significant.

### 3.6. Diagnostic Value of MUC5AC

The diagnostic value of MUC5AC glycoforms was tested to distinguish PDA from non-PDA tissues. Later we tested the ability of MUC5AC to diagnose NpD (PDA and NET) vs. Non-NpD. The ML were excluded for these analyses.

The results of PDA vs. non-PDA are discussed in Table 2. The SN was highest when both glycoforms were used (71%), and the drop was >5% with IM-alone; it was worse with MM-alone or location-specific MM. The SP was 100% when MM (any location) was used and dropped to 93% when IM was used alone or in combination with MM. The *p*-value was always significant.

The results of NpD vs. Non-NpD are discussed in Table 3. The trend was the same as that of the PDA vs. non-PDA. The SN was highest when MM and IM were used (59%) for staining tissues. It dropped slightly to 55% when IM-alone was tested. When it was a location-specific MM expression, the SN worsened with MM detection from the cytoplasm to the extracellular region (Cy > Ap > Ec). SP was 100% irrespective of the glycoform (MM vs. IM) or location of the MM (Ap vs. Cy vs. Ec) used, and the *p*-value was always significant (<0.05).

## 4. Discussion

Classification of MUC5AC into MM and IM glycoforms was not well studied, so we attempted it in commercially available TMA with various NpDs and non-NpPDs. It gave us an opportunity to study its impact on pathological features such as stage, histology, grade of differentiation, node positivity, and size of the PT (T-stage). To our knowledge, this is the first study reviewing MUC5AC expression in pancreatic NETs.

Our study shows MUC5AC (combination of IM and MM) has low SN (59%) and high SP (96%) to diagnose PDA from non-PDA tissues (NET + non-NpD). MUC5AC was not expressed in normal and pancreatitis tissues, and it was detected in PDA which is consistent with prior studies [11,13,15]. Inflammatory responses do not seem to trigger MUC5AC production. We may infer that MUC5AC has a role in the neoplastic process (PDA or NET) but is not necessary based on its prevalence among NpD tissues in TMA [16]. We do not have enough information to know the impact of IM detected in the two NETs that were positive for it in the TMA compared to the rest of them. Only PDA tissues (PT and ML) expressed MM. Hence, MM could be explored as a biomarker to distinguish PDA from pancreatic NET.

The general hypothesis that IM is a precursor for MM is supported by the dual expression of both isoforms in most MM-positive cases, especially with Ec MM. High CLH2 H-score in MM-positive tumors (vs. negative tumors) indicates that the intensity of IM is an important factor while evaluating its clinical significance instead of the traditional positive and negative analysis. Factors that trigger heavy glycosylation and transformation of IM to MM are yet to be discovered. The heterogeneity of MUC5AC expression in tumors (MM > IM) as seen in our study is confirmed by other studies, and it is interesting to note that more than half of patients positive for MM could have been missed if just one core was used for examination [17]. This raises a question of whether this could be the reason for inconclusive results noted in previous TMA-based studies with just one core [11,13,16].

Among the PTs, IM detection (66%) was similar to other studies in the literature, but MM expression was low (47%) [10,11,17,18]. A study published two decades ago that used different MM clones, 21-M1 and Nd-2, showed a prevalence of 92% and 68%, respectively [17]. Prior studies indicate that Nd-2 clone binds to same epitope as 45M1 [19]. Extrapolating the data from that study (using Nd-2 as a surrogate to 45M1), expression of 45M1 in this study is lower than expected. Early-stage and G1–G2 tumors tend to express MUC5AC more. Other clinicopathologic characteristics available to us such as node positivity and size (T-2 vs. T3-4) were not significantly influenced with MUC5AC detection.

The sample size in our study is too small to draw definitive conclusions on MM expression in ML. Prior studies reported IM detection in liver metastatic lesions, but they were negative in our study [13]. Omental lesions in that study were positive for IM whereas it was positive for both glycoforms in our study. Larger studies using matched PT and ML lesions are needed to further investigate the role (if any) of MUC5AC in metastasis as preclinical evidence suggests MM might have a role in it [7]. Among the PDAs positive for MM, we could not establish a definite correlation between Ap and Cy or Ap and Ec or Cy and Ec. This variability in the localization of MM (Ap vs. Cy vs. Ec) is another wrinkle to assessing the clinical significance of MM in PDA that needs to be addressed in future studies.

This study with its results consistent with available evidence supports further pursing the use of MM since its diagnostic value in the current clinical practice is unclear. We can argue staining biopsies/cytology specimens with IM and MM could help in diagnosing PDA if there is a concern for malignant transformation of IPMN or cysts with inconclusive pathology results. Prior preclinical studies suggested epigenetic silencing (by methylation) of a MUC5AC gene promoter is one of the mechanisms suppressing its constitutive expression in normal tissues [20,21,22]. However, the triggers for its production are not well established, and our study was not designed to investigate it. The association between MUC5AC detection and other genetic alternations often linked with carcinogenesis such as activating KRAS mutations needs to be examined in larger studies. We used pancreatic TMAs, which allow for screening a large number of cases in a fast and cost-effective manner. However, examination of limited tissue may not be entirely representative of the tumor, as PDAs are proven to be heterogeneous tumors. There were not enough MLs to understand the difference in IM and MM expression patterns between PT and ML. We did not have the required clinical (recurrence rates, treatment response, and survival) and genomic information (mutation profiling) to draw clear conclusions on the impact of MUC5AC expression in PDAs.

Furthermore, we did not have the required clinical (recurrence rates, treatment response, and survival) and genomic information to draw clear conclusions on the triggers or impact of MM expression in PDAs. Another major limitation of the study is that only PDA and NET are studied for the expression of MUC5AC, and the TMAs used did not include other neoplasms of the pancreas such as pancreatic intraepithelial neoplasia (PanIN), IPMN or cystic neoplasms, acinar cell carcinoma, and pancreatic neuroendocrine carcinomas. It would be interesting to see the expression of MUC5AC in mucin-producing cystic neoplasms of the pancreas, including intraductal papillary mucinous neoplasms, and further examine the expression in the malignant transformation of these neoplasms. We used pancreatic TMAs, which allow for screening a large number of cases in a fast and cost-effective manner. However, examination of limited tissue may not be entirely representative of the tumor, as PDAs are proven to be heterogeneous tumors.

## 5. Conclusions

In our preliminary study with patient tissue samples in commercially available TMA, we have shown that there is selective expression of MUC5AC isoforms (IM vs. MM) in the neoplastic tissue, especially in PDA. This study provides more evidence to the known diagnostic value of IM and, for the first time, highlights the benefit of adding MM to it. It confirms differential expression of IM (Cy-only) and MM (Cy, Ap, and Ec) and offers a basis to further explore the role of MUC5AC as a potential diagnostic, prognostic, and predictive biomarker in PDA.

## 6. Patents

The Ohio State University is currently pursuing patent protection for the research discussed in this publication.

## Figures and Tables

**Figure 1 cancers-15-04832-f001:**
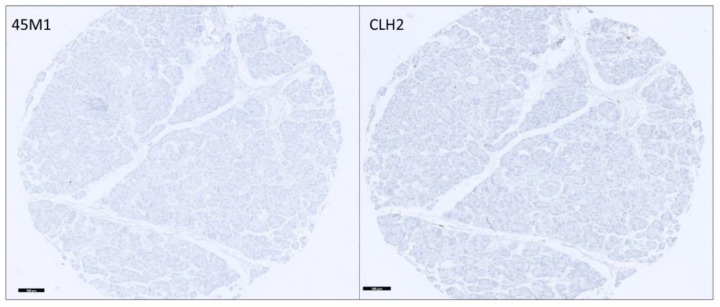
Normal pancreatic tissue with no staining with 45M1 and CLH2 clones (100× magnification).

**Figure 2 cancers-15-04832-f002:**
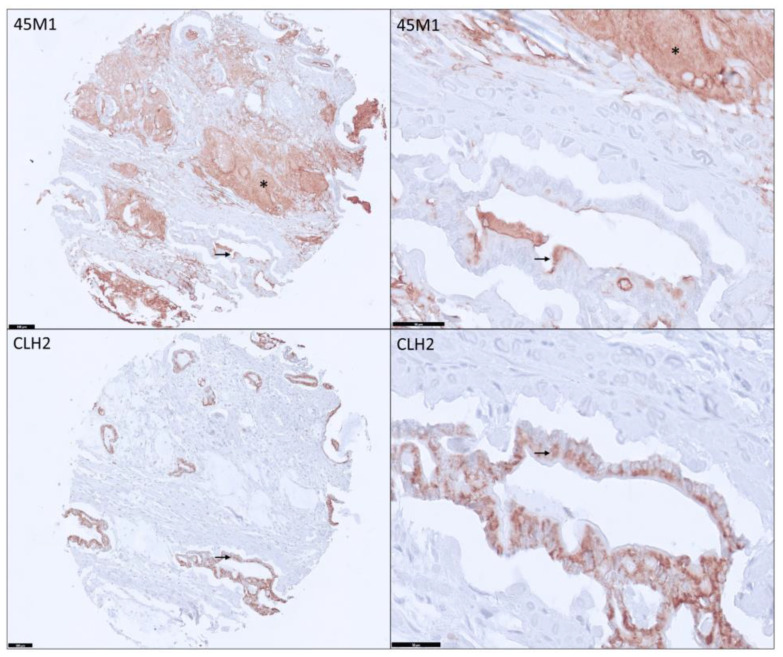
Pancreatic ductal adenocarcinoma showing patchy apical stain in the tumor cells (black arrow) and strong extracellular staining (*) for 45M1 and cytoplasmic expression for CLH2 (black arrow) without any extracellular staining (100× and 400× magnification).

**Figure 3 cancers-15-04832-f003:**
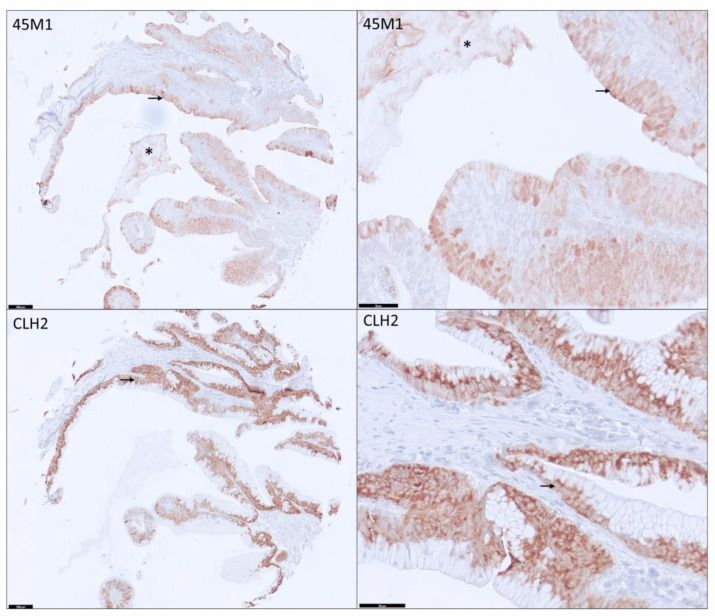
Diffuse uniform expression of 45M1 and CLH2 in pancreatic ductal adenocarcinoma core with apical expression (black arrow) and some weak extracellular mucin (*) for 45M1 and only cytoplasmic expression for CLH2 (100× and 400× magnification).

**Figure 4 cancers-15-04832-f004:**
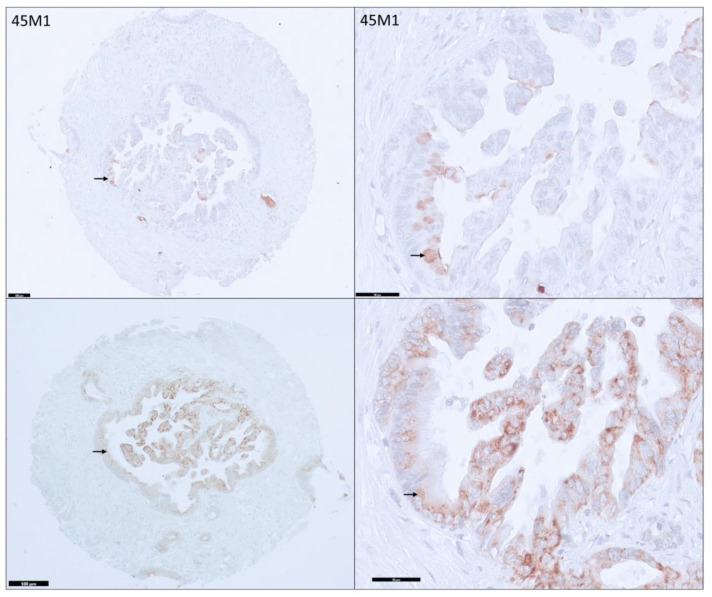
Variable expression for 45 M1 and CLH2 in a pancreatic adenocarcinoma core. Only focal apical expression (black arrow) is noted for 45 M1, whereas diffuse cytoplasmic expression (black arrow) is noted with CLH2 (100× and 400× magnification).

**Table 1 cancers-15-04832-t001:** The patient population and staining results of TMA with mature and immature MUC5AC.

	N	MM ± IM	MM-Positive	MM-Only	Ap/Cy/Ec	IM-Only	MM + IM	All Negative
Neoplastic	54	30	19	2	12/16/9	11	17	24
• PDA	43	28	19	2	12/16/9	9	17	15
■ Primary tumor ■ Metastatic tumor	38 5	27 1	18 1	2 0	11/15/9 1/1/0	9 0	16 1	11 4
• NET	11	2	0	0	0	2	0	9
Non-neoplastic	42	0	0	0	0	0	0	0
• Normal adjacent	24	0	0	0	0	0	0	0
• Normal	10	0	0	0	0	0	0	0
• Pancreatitis	8	0	0	0	0	0	0	0

MM—mature MUC5AC, IM—immature MUC5AC, NET—neuroendocrine tumors, PDA—pancreatic ductal adenocarcinoma, Ap—apical expression, Cy—cytoplasmic expression, Ec—extracellular expression.

**Table 2 cancers-15-04832-t002:** Diagnostic value of MUC5AC glycoforms to diagnose PDA from non-PDA pancreatic lesions.

MUC5AC Glycoform	Sensitivity (95% CI)	Specificity (95% CI)	PPV (95% CI)	NPV (95% CI)	Accuracy (95% CI)	*p* Value
MM or IM	71.05% (54.1%–84.58%)	96.23% (87.02%–99.54%)	93.10% (77.23%–99.15%)	82.23% (70.47%–90.8%)	85.71% (76.81%–92.17%)	<0.0001
MM + IM	42.11% (26.31%–59.18%)	100% (93.28%–100.00%)	100% (79.41%–100.00%)	70.67% (59.02%–80.62%)	75.82% (65.72%–84.19%)	<0.0001
MM positive	47.37% (30.98%–64.18%)	100% (93.28%–100%)	100% (81.47%–100%)	72.60% (60.91%–82.39%)	78.02% (68.12%–86.03%)	<0.0001
IM positive	65.79% (48.65%–80.37%)	96.23% (87.02%–99.54%)	92.59% (75.71%–99.09%)	79.69% (67.77%–88.72%)	83.52% (74.27%–90.47%)	<0.0001
MM Ap	28.95% (15.42%–45.90%)	100% (93.28%–100%)	100% (71.51%–100%)	66.25% (54.81%–76.45%)	70.33% (59.84%–79.45%)	<0.0001
MM Cy	39.47% (24.04%–56.61%)	100.00% (93.28% to 100%)	100% (78.20%–100.00%)	69.74% (58.13%–79.75%)	74.73% (64.53%–83.25%)	<0.0001
MM Ec	23.68% (11.44%–40.24%)	100 (93.28%–100%)	100% (66.37%–100%)	64.63% (53.30%–74.88%)	68.13% (57.53%–77.51%)	0.0002

MM—mature MUC5AC, IM—immature MUC5AC, Ap—apical expression, Cy—cytoplasmic expression, Ec—extracellular expression, PPV—positive predictive value, NPV—negative predictive value.

**Table 3 cancers-15-04832-t003:** Diagnostic value of MUC5AC glycoforms to diagnose neoplastic from non-neoplastic pancreatic lesions.

MUC5AC Glycoform	Sensitivity (95% CI)	Specificity (95% CI)	PPV (95% CI)	NPV (95% CI)	Accuracy (95% CI)	*p* Value
MM or IM	59.2% (44.21%—73%)	100% (91.59%–100%)	100% (88.06%-100%)	67.7% (54.66%–79.06%)	78.1% (68.12%–86.03%)	<0.0001
MM+ IM	32.65% (19.95%–47.54%)	100% (91.59%–100%)	100% (79.41%–100%)	56% (44.06%–64.36%)	63.74% (52.99%–73.56%)	<0.0001
MM positive	36.73% (23.42%–51.71%)	100% (91.59%–100%)	100% (81.47%–100%)	57.53% (45.41%–69.03%)	65.93% (55.25%–75.55%)	<0.0001
IM positive	55.1% (40.23%–69.33%)	100% (91.59%–100%)	100% (87.23–100%	65.625% (52.7%–77.05%)	75.82% (65.72%–84.19%)	<0.0001
MM Ap	22.45% (11.77%–36.62%)	100% (91.59%–100%)	100% (71.51%–100%)	52.5 (41.02%–63.79%)	58.2% (47.43%–68.5%)	0.0011
MM Cy	30.61% (18.25%–45.42%)	100.00% (91.59%–100%)	100% (78.20%–100%)	55.26% (43.41%–66.69%)	62.64% (51.87%–72.56%)	<0.0001
MM Ec	18.36% (8.76%–32.02%)	100% (91.59%–100%)	100% (66.37%–100%)	51.22% (39.92%–62.42%)	56.04% (45.25%–66.44%)	0.0034

MM—mature MUC5AC, IM—immature MUC5AC, Ap—apical expression, Cy—cytoplasmic expression, Ec—extracellular expression, PPV—positive predictive value, NPV—negative predictive value.

## Data Availability

We used commercially available TMA. The data is available on manufacturer’s website (https://www.tissuearray.com/PA2081c).

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
