# Peer review of "Differential Expression and Diagnostic Value of MUC5AC Glycoforms in Pancreatic Ductal Adenocarcinoma"

_cancers, 2023, doi:10.3390/cancers15194832_

Round 1
Reviewer 1 Report
In the present manuscript, authors have evaluated the expression of two different glycosylated forms of MUC5AC. According to their study, highly glycosylated form of MUC5AC exhibits more sensitivity and specificity in pancreatic cancer detection. Its an interesting study but need to be presented in a better way. I have the following concerns:
1-Authors need to improve the abstract, it should be conclusive and abbreviation should not be used without prior description.
2-Figure presentation is not appropriate, Normal and tumor should be presented together, all the images should have the size bar.
3-Results needs to be described in more quantitative manner.
English language if fine, just need to check for typos.
Author Response
- Authors need to improve the abstract, it should be conclusive and abbreviation should not be used without prior description.
-
- We appreciate the feedback. We added following lines (36-38) to the end of the abstract, The combination of MUC5AC glycoforms has high SP and reasonable SN to diagnose PDA. They can potentially be a reliable diagnostic marker alone or in combination with other biomarkers and should be investigated further in more extensive studies.
- Figure presentation is not appropriate, Normal and tumor should be presented together, all the images should have the size bar.
-
- We appreciate the feedback. We changed the figures (added Figure 4). Added the size bar. To keep the flow of the manuscript we separate normal and tumor figures.
- Results needs to be described in more quantitative manner.
-
- We appreciate the feedback. The primary aim of our study was to examine the differential expression (detection) of major glycoforms in neoplastic, normal, and benign pancreatic tissues. We did our best to report the results in quantitative results using H-scores.
Reviewer 2 Report
Dear authors
The excessive number of unnecessary abbreviations makes the reading of the manuscript slow and unpleasant. Please, change this situation.
Many abbreviations are not explained. for example in the abstract you speak of TMA. It happens that I do not know what TMA is. I suppose is tissue micro array but you have to be clear.
Can you please tell me what SN and SP mean. I suppose it is sensitivity and specificity. The average reader would not know.
You say
There is strong preclinical evidence suggesting MM offers gemcitabine resistance to PDA that escaped the attention as none of the prior studies were designed to examine it [7]
However, this strong preclinical evidence is based on a self citation. Please, add other authors references that support this idea.
When you describe the specimens, you have to clarify if it is acute or chronic pancreatitis. I suppose it is chronic, but you have to be clear.
The quality of figure 1 is very poor, It would be nice if you remove that light blue ground glass covering the figure.
I recommend to do the same in figures 2 and 3.
All the histological figures have very low magnification. I think you have to add a figure with higher magnification showing the different distribution of staining in the cells and extracellular substance.
Tables 2 and 3 show that MUC5AC has very low sensitivity although specificity is high. This point deserves to be included in the discussion.
An important issue omitted in your research is about the driver gene. This leaves out the possibility to determine if KRAS mutation has any influence on the presence or absence of MUC5AC.Same story with other possible genes.
Is MUC5AC gene mutated...we do not know and you did not test this issue.
English is acceptable
Author Response
- The excessive number of unnecessary abbreviations makes the reading of the manuscript slow and unpleasant. Please, change this situation.
-
- We appreciate the feedback. We revised the manuscript and removed abbreviations for many terms including normal (N), pancreatitis (P), normal adjacent (NAJ), undifferentiated carcinoma (UC), and adenosquamous (AS).
- We used only essential ones such as for, mature (MM) and immature (IM) MUC5AC, Sensitivity (SN), specificity (SP), tissue micro array (TMA), neoplastic (NpD) and non-neoplastic (non-NpD), pancreatic adenocarcinoma (PDA) and non-PDA, and neuroendocrine tumor (NET).
- Many abbreviations are not explained. for example in the abstract you speak of TMA. It happens that I do not know what TMA is. I suppose is tissue micro array but you have to be clear.
-
- We appreciate the feedback. Clarified TMA in the line 27.
- Can you please tell me what SN and SP mean. I suppose it is sensitivity and specificity. The average reader would not know.
-
- We appreciate the feedback. Clarified these abbreviations as sensitivity and specificity in the abstract (lines 32-33).
- You say There is strong preclinical evidence suggesting MM offers gemcitabine resistance to PDA that escaped the attention as none of the prior studies were designed to examine it [7]. However, this strong preclinical evidence is based on a self-citation. Please, add other authors references that support this idea.
-
- We appreciate the feedback. We added following lines (68-74) in the introduction, There is strong preclinical evidence suggesting MM offers gemcitabine resistance to PDA was first reported by Krishn et al. that escaped the attention as none of the prior studies were designed to examine it [12]. The authors showed that gemcitabine-sensitivity im-proved with the inhibition of MUC5AC in the pancreatic cell lines. We elaborated on it in a paper published earlier this year [7]. Similarly, the relationship between IM and MM, and its effect on carcinogenesis, local invasion, metastasis, and treatment response was not investigated.
- When you describe the specimens, you have to clarify if it is acute or chronic pancreatitis. I suppose it is chronic, but you have to be clear.
-
- We appreciate the feedback. We have One acute and 7 chronic pancreatitis. We added this to section 2.1 (lines 91-92).
- The quality of figure 1 is very poor, It would be nice if you remove that light blue ground glass covering the figure. I recommend to do the same in figures 2 and 3. All the histological figures have very low magnification. I think you have to add a figure with higher magnification showing the different distribution of staining in the cells and extracellular substance.
-
- We appreciate the feedback. We changed the figures. Added high magnification figures.
- Tables 2 and 3 show that MUC5AC has very low sensitivity although specificity is high. This point deserves to be included in the discussion.
-
- We appreciate the feedback. We added following lines (253-255) to the discussion, Our study shows MUC5AC (combination of IM and MM) has low SN (59%) and high SP (96%) to diagnose PDA from non-PDA tissues (NET + non-NpD). MUC5AC was not expressed in normal and pancreatitis tissues and detected in PDA which is consistent with prior studies [11,13,15].
- An important issue omitted in your research is about the driver gene. This leaves out the possibility to determine if KRAS mutation has any influence on the presence or absence of MUC5AC.Same story with other possible genes. Is MUC5AC gene mutated...we do not know and you did not test this issue.
-
- We appreciate the feedback. We added following lines (294-299) to the discussion, Prior preclinical studies suggested epigenetic silencing (by methylation) of MUC5AC gene promoter is one of the mechanisms suppressing its constitutive expression in normal tis-sues [20-22]. However, the triggers for its production are not well established, and our study was not designed to investigate it. The association between MUC5AC detection and other genetic alternations often linked with carcinogenesis such as activating KRAS mutations needs to be examined in larger studies.
Round 2
Reviewer 2 Report
No further comments